# Computational design of Periplasmic binding protein biosensors guided by molecular dynamics

Jack M. O'Shea[1,2]*, Peter Doerner[1], Annis Richardson[1], Christopher W. Wood[1]*

1 School of Biological Sciences, University of Edinburgh, Edinburgh, United Kingdom, 2 School of Natural Sciences, Technical University of Munich, Center for Functional Protein Assemblies (CPA), Garching, Germany

* jack.oshea@tum.de (JMOS); chris.wood@ed.ac.uk (CWW)

**Data Availability Statement:** All code and data required to recreate all the analysis in this paper is available on GitHub under a permissive MIT

## Abstract

Periplasmic binding proteins (PBPs) are bacterial proteins commonly used as scaffolds for substrate-detecting biosensors. In these biosensors, effector proteins (for example fluorescent proteins) are inserted into a PBP such that the effector protein's output changes upon PBP-substate binding. The insertion site is often determined by comparison of PBP *apo*/*holo* crystal structures, but random insertion libraries have shown that this can miss the best sites. Here, we present a PBP biosensor design method based on residue contact analysis from molecular dynamics. This computational method identifies the best previously known insertion sites in the maltose binding PBP, and suggests further previously unknown sites. We experimentally characterise fluorescent protein insertions at these new sites, finding they too give functional biosensors. Furthermore, our method is sufficiently flexible to both suggest insertion sites compatible with a variety of effector proteins, and be applied to binding proteins beyond PBPs.

## Author summary

"Biosensors" are microscopic tools that can detect specific molecules of interest and are made of biological building blocks, such as proteins. Upon coming into contact with their target molecule, such as a marker of a specific disease, biosensors change shape and their properties are altered. For example, upon contact with a disease marker, a biosensor could glow brightly to work as a diagnostic, or perhaps it could produce a chemical signal that can be detected by the immune system. Understanding how biosensors change shape in response to their target is key to developing more biosensors with complex responses. In this paper, we use computer simulations of a biosensor component to understand what parameters are important for biosensor design. We use these parameters to generate new biosensors that respond to their target molecule by producing light, but put forward a case that the lesson learned are generalisable enough to inform more complex sensor outputs as well.

License: https://github.com/wells-wood-research/oshea-j-wood-c-pbp-design-2023.

**Funding:** This work was supported by a BBSRC Engineering Biology Breakthrough Award (BB/W013320/1 to CWW). The funders had no role in study design, data collection and analysis, decision to publish, or preparation of the manuscript.

# Introduction

Protein-based biosensors are important tools for research in biochemistry and medicine. Being genetically encodable, they can monitor real-time *in vivo* conditions and can be optimised by directed evolution. The minimum requirement for a biosensor is that is possess an input-sensing "detector" domain and an output-generating "effector" domain. Modularity of detector and effector domains makes design of new sensors much simpler, which is why the periplasmic binding protein scheme of protein-based biosensors has been so successful.

Periplasmic binding proteins (PBPs) are a large family of bacterial proteins that scavenge and sense nutrients, and so are very well suited to detecting small molecules [1,2]. They have evolved to bind a wide range of metabolically important ligands with high affinity and specificity, and new ligand specificities have been engineered by directed evolution [3,4] and computational methods [5,6]. Upon substrate binding they undergo significant conformational change (Fig 1A). For this reason their *apo* (without ligand) and *holo* (ligand-bound) states to be distinguished by receptors and thereby regulate many processes downstream in bacterial cells [7,8]. Structurally, PBPs possess two distinct folded lobes that are spanned by flexible inter-lobe loops, and upon substrate binding these lobes close around the substrate with a Venus flytrap- or hinge-like movement. In other words, when PBPs are *apo* (without ligand) they are most likely in an "open" conformation, and when they are *holo* (ligand-bound) they are in "closed" conformation (Fig 1A).

In PBP biosensors, the PBP's structural changes upon substrate binding are physically transmitted to a fused effector protein so as to measurably change the effector's output. Sensors with outputs such as fluorescence [9–16], transcriptional repression [17,18], and enzymatic activity [19–22] have been generated by inserting different effector proteins into PBP sequences, demonstrating the modularity of this approach. Typically, effectors are circularly permuted: their sequence is rearranged such that their N and C termini are very close to each

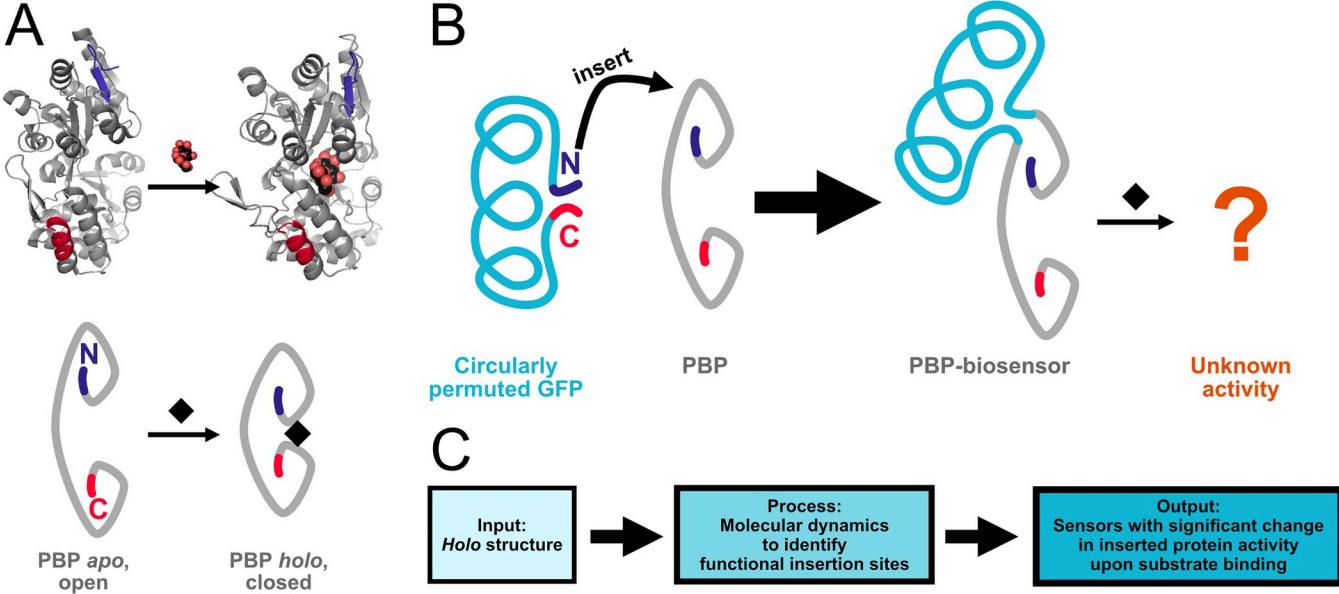

**Fig 1. Schematics explaining design of PBP-biosensors. A)** Crystal structures of *apo* (1ANF) and *holo* (1OMP) states of periplasmic binding proteins (PBPs), with schematic representations underneath. N and C termini are highlighted in blue and red respectively. **B)** Circularly permuted GFP has its N and C termini close together, so it can be inserted into PBPs to generate biosensors, but it is difficult to predict sensor activity from insertion site. **C)** Our method takes *holo* PBP structure as input, and using molecular dynamics simulations can provide insertion sites to generate functional biosensors.

other, ideal for insertion into the PBP sequence [2,23,24]. However, despite many examples of PBP biosensors, their design is impeded by our limited ability to identify sites in PBPs where effectors can be inserted to produce active sensors (Fig 1B), which we term "the identification of functional insertion sites".

Functional insertion sites were first rationally identified by the comparison of *apo* and *holo* crystal structures [10]. Marvin *et al.* showed that regions of greatest change in backbone dihedral torsion angles between the *apo*-open and *holo*-closed states of the maltose binding protein were associated with production of viable sensors upon insertion of circularly permuted GFP (cpGFP). This method has been successfully applied to other PBPs, either by referring to *apo*/*holo* crystal structures or by homology modelling of known successful insertion sites onto the new PBP [9,11,25]. However, change in crystal structure backbone torsion angle can be an unreliable indicator of good insertion sites: sites of high change can be nonfunctional [10] and sites of low change can be excellent [12,18]. Clearly crystal structure backbone torsion does not explain all factors necessary for functional insertion into PBPs. Alternatively, functional insertion sites have been identified by screens of random insertion libraries [12,19,21], but these methods are labour intensive, difficult to get full insertion coverage with, and do not explain why the identified insertion sites are functional.

In order to address these current shortcomings in PBP-based biosensor design, we reasoned that molecular dynamics (MD) could contain data that discern functional insertion sites better than crystal structure backbone torsion. We applied structural modelling, simulation, and analysis to the task (Fig 1C) using the well-researched maltose binding protein (MBP) as a model subject. In simulations, requiring only the *holo* structure as a starting point, we captured MBP's transition from closed to open state. We found that regions with substantial changes in residue contacts were correlated with the previously identified functional insertion sites, but the analysis also identified additional sites. From these additional sites, we generated new viable MBP sensors with a range of properties. We therefore propose that "change in residue contacts" is a more reliable indicator for functional insertion sites than "crystal structure backbone torsion". This represents a significant advance in our understanding of the design parameters of PBP-based biosensors and demonstrates that computational analysis can greatly reduce barriers in the novel design of such sensors.

## Results and discussion

Previous successful design of PBP biosensors has relied on inserting effector proteins into PBPs at regions of putative substrate-induced conformational change. We reasoned that functional insertion sites could be identified by using molecular dynamics simulations that capture such conformational change. As a test subject for this approach, we chose the maltose binding protein (MBP). Two previous studies provide screening data of large libraries of MBP with randomly inserted effector proteins. Nadler *et al.* randomly inserted a cpGPF, observing the greatest maltose-induced fluorescence changes from insertions at residues 169–171, followed by residues 355–348 [12]. Younger *et al.* inserted a zinc-finger transcription factor finding that insertion at position 335 produced the greatest maltose-induced change in transcription [18]. Agreement in 335 as a good insertion site supports 355's significance and shows that some insertion sites can be used with many different effectors. Therefore, the two regions of interest for our study are 169–171 and 335–348, and our aim was to find properties in molecular dynamics that distinguish these regions from all others.

We performed simulations of the *holo* / "closed" state using a crystal structure (PDB: 1ANF) as a starting point [26]. 10 x 100 ns simulations were conducted (Fig 2A and 2B). Simulations to sample the open state were also performed from a starting point of 1ANF, except

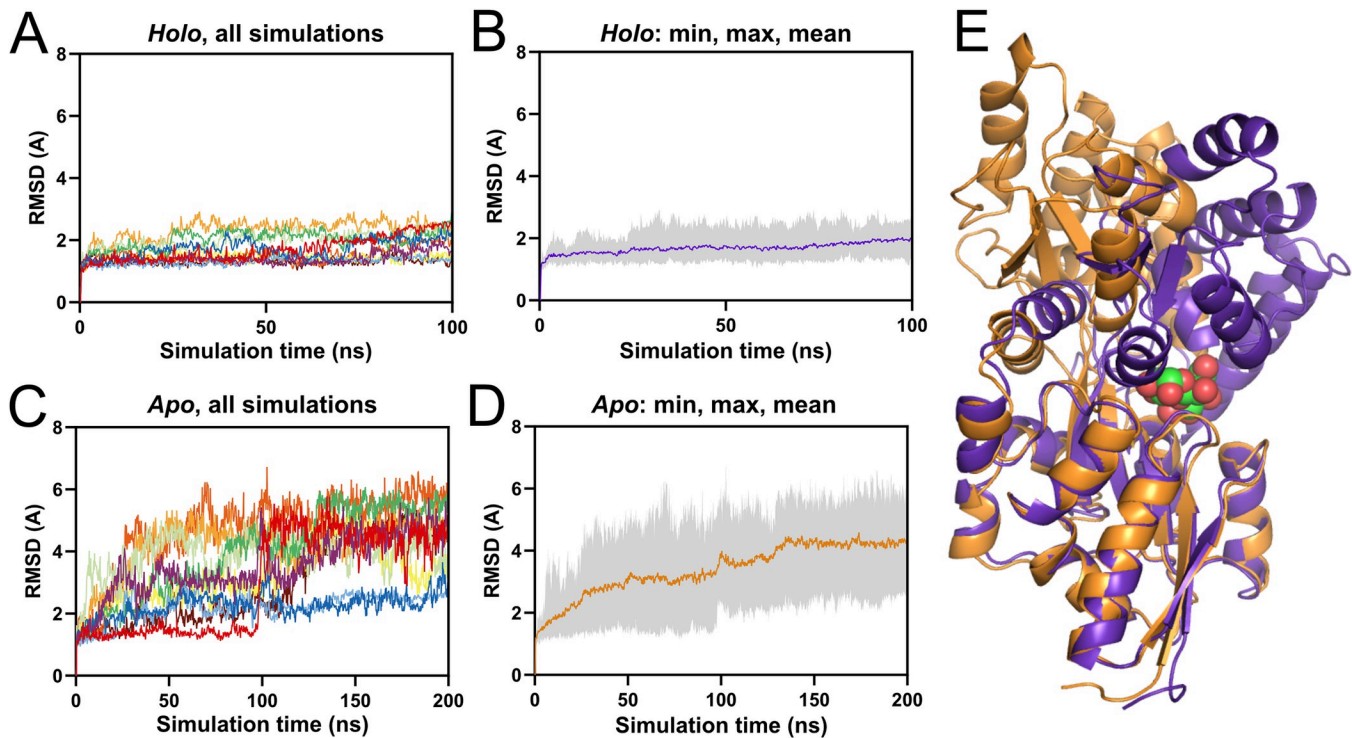

**Fig 2. RMSD of molecular dynamics simulations of Maltose Binding Protein (MBP) in *apo* and *holo* forms. A, B:** *Holo* simulations were generated from the starting point of the MBP/maltose complex crystal structure. **A)** Individual traces for 10x100ns MBP *holo* simulations. **B)** The mean, minimum, and maximum RMSD values for the MBP *holo* simulations. **C, D:** *A*po simulations were generated from the starting point of the MBP/maltose complex crystal structure with maltose removed. **C)** Individual traces for 10x200ns MBP apo simulations. **D)** The mean, minimum, and maximum RMSD values for the MBP apo simulations. **E)** Final poses of MBP in representative *holo*/closed (purple) and *apo*/open (orange) simulations, chosen for RMSD close to the mean-of-means for each condition. Maltose is shown in the *holo* structure in green.

with maltose manually removed, and then simulations were run long enough for the protein to transition from this closed starting point to the open state. This was a self-imposed restriction as we were aiming to produce a method that did not require crystal structures of both closed and open states (although both do exist for MBP). We ran 10 x 200 ns of these *apo* simulations, observing much higher RMSDs in most of the simulations (Fig 2C and 2D). Watching these simulations, we could see that the increase in RMSD came solely from a separation of the N- and C-lobes via a hinging motion at the inter-lobe loops (Fig 2E). Among the closed-state simulations we saw that none of the mean RMSDs exceed 2.5 A from the starting conformation. We therefore chose to define simulations as "closed" when mean RMSD for the last 50 ns is < 2.5 A and "open" when it is > 2.5 A. By this definition, we saw the open state in 8/10 of the *apo* simulations. RMSD by comparison to the *apo* crystal structure (PDB: 1OMP) shows that the *apo* simulations evolve to be closer to the *apo* crystal structure, settling at a mean RMSD of ~3 A (S1 Fig).

The first metrics we investigated were changes in backbone dihedral torsion (ΔDihe-MD) and root mean squared fluctuation (ΔRMSF-MD) between closed and open simulations. For ΔDihe at residue *i*, the dihedral angle between alpha-carbon atoms c-$\alpha_{i-1}$, c-$\alpha_i$, c-$\alpha_{i+1}$, and c-$\alpha_{i+2}$ is taken [10]. Between the MBP 1OMP (*holo*) and 1ANF (*apo*) crystal structures, residue 175 has the greatest ΔDihe [10], although Nadler and colleagues' random insertion at residue 171 was found to produce a better sensor [12]. ΔRMSF was chosen to investigate regions of flexibility and conformational change within the protein. ΔDihe and ΔRMSF were calculated

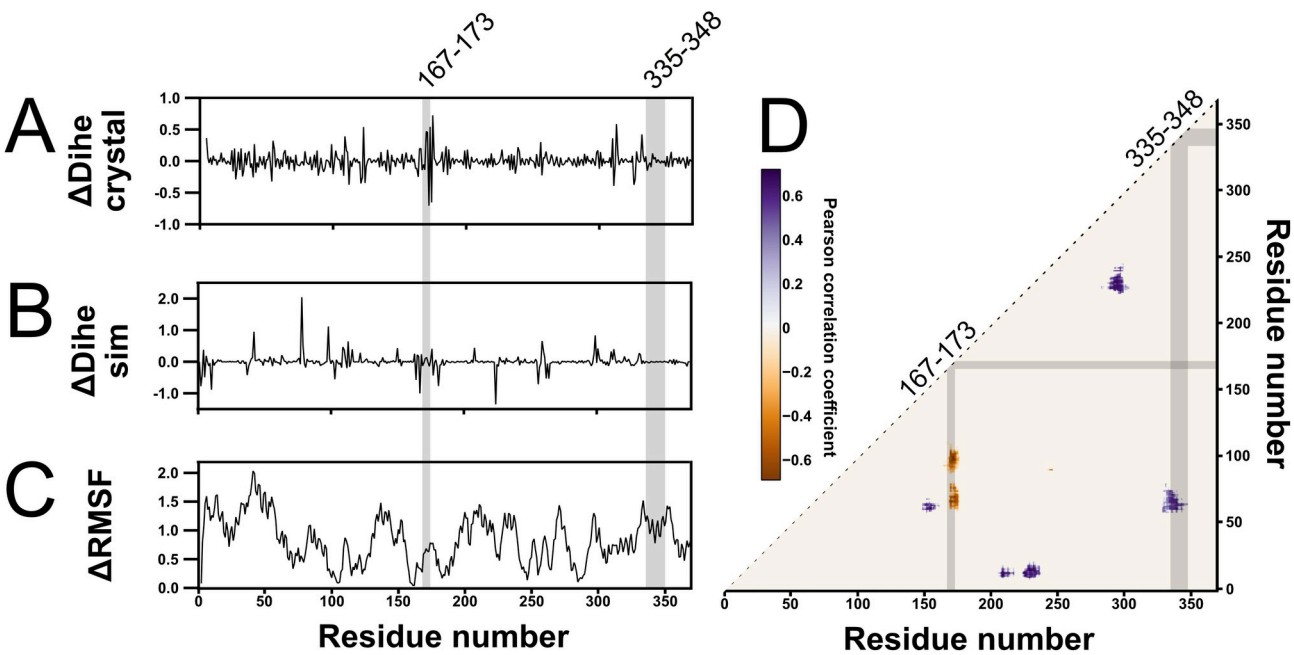

**Fig 3. A comparison of metrics attempting to identify regions of MBP primary sequence known to be receptive to effector protein insertion with highlighted regions of interest.** Across all figures, the residue ranges 167–173 and 335–348 are highlighted. 167–173 contains the region of interest 169–171, with flanking residues also highlighted for ease of visualisation. **A)** The change in backbone dihedral angle between *holo* and *apo* crystal structures (PDB 1ANF, 1OMP respectively). **B)** Change in backbone dihedral angle between *holo* and *apo* molecular dynamics simulations. Neither regions 167–173 nor 335–348 contain significant values for this metric. **C)** Difference in RMSF of each residue between *holo* and *apo* molecular dynamics simulations. **D)** Pearson correlation coefficient metric for all *apo* MBP simulations where the closed-to-open state transition was observed, filtered to only show peaks in the top percentile for mean, median, and maximum pixel value. Negative Pearson correlation coefficient indicates residues that become closer to each other throughout the closed-to-open transition, and positive values closer.

for each residue for each of the final 50 ns of each closed-state simulation and for each of the simulations that reached the open-state, and the mean value in the open-state was subtracted from that of the closed. We compared these to ΔDihe of the crystal structure (ΔDihe-Crys). We hoped that at least one of ΔDihe-MD and ΔRMSF-MD would highlight our regions of interest 169–171 and 335–348.

Unfortunately, ΔDihe-MD and ΔRMSF were worse at highlighting our regions of interest than ΔDihe-Crys (Fig 3AC). Residues in the region of interest (ROI) 169–171 have some of the largest ΔDihe-Crys values, with 170 and 171 ranking 10th and 11th. ΔDihe-MD and ΔRMSF do not discern residues of this range at all. For the 335–348 range, residues did not stand apart for any measurements in MD or crystal structure. This shows that crystal structures can discern some good insertion sites (169–171) but not all (335–348). This initial analysis of the MD trajectory provided worse sites than using the static crystal structure alone, indicating that these metrics were not sensitive to the kinds of motions required to generate functional sensors.

In order to identify more subtle structural dynamics, we applied the program CONAN (CONtact ANalysis) [27] to our MD data. CONAN produces many metrics on the changes in interactions of a structure throughout a simulation. We submitted our *apo*-MBP simulations to CONAN, thereby providing it dynamic data on the close-to-open transition. Of interest to us was the Pearson correlation coefficient matrix output. This matrix would show us regions that experience changes in contacts between closed and open states, so may be involved in the changes in conformation that distinguish functional insertion sites. We defined a "contact" between residues as beginning when the closest-atom distance is less than 0.5 A. A contact is only terminated if the closest-atom distance exceeds 0.8 A (so that momentary losses in contact

                      

are not overly interpreted). The resulting Pearson correlation coefficient matrix has one "pixel" for each residue pair. Positive values indicate interactions that occur early in the simulation (when MBP is closed) and negative values indicate interactions that occur late in the simulation (when MBP is open). To find the most significant hotspots of the Pearson correlation coefficient matrix, first an average-Pearson matrix was produced, with each pixel value being the mean of that pixel's value across the Pearson matrices from each individual simulation (S2 Fig). Pixels were then sorted into "clusters", defined as contiguous areas of positive values (peaks) or negative values (valleys). For each cluster, mean pixel value, median pixel value, and maximum absolute pixel value were calculated. Finally, peaks and valleys were filtered for those that were in the top percentile of mean, median, and max absolute values simultaneously (peaks and valleys ranked separately).

In the resulting mean-filtered Pearson matrix (Fig 3D), it is clear that our regions of interest 169–171 and 335–348 experience some of the most substantial changes in residue contacts during MBP opening in our MD. ROI 169–171 aligns with an area of negative Pearson space, indicating contacts that only occur in the open state. ROI 335–348 aligns with an area of positive Pearson space, indicating contacts that only occur in the closed state. The Pearson matrix calculated from the comparison of the MBP closed and open crystal structures (S3 Fig) has too much noise to discern any ROIs, demonstrating the necessity of MD-generated data. These data provide the first unifying feature of the known insertion hotspots 169–171 and 335–348: changes in contacting residues between closed and open state.

It is particularly of note that this metric captures ROI 335–348, which is not identified by comparison of crystal structures (Fig 3A–3C) but is identified by all random insertion studies to date [12,18]. These studies randomly inserted cpGFP and a zinc-finger transcription factor respectively and both identified residue 335, showing that functional insertion sites can be agnostic to the effector placed there. Another rational insertion study inserted beta-lactamase at 164 to achieve maltose-dependent antibiotic resistance [19], which is at the edge of the 170-centred CONAN hotspot. Change in residue contacts seems therefore to be a consistently important factor for a variety of sensor outputs, namely fluorescence, transcription, and catalysis. From this perspective, we propose that Pearson hotspots therefore correspond to regions where an inserted effector will experience significant changes in residue environment upon substrate binding, and that these changes are likely to significantly modulate effector output.

To validate this rationale for identifying good MBP insertion sites, we used the Pearson data to guide the design of new maltose sensors. We decided to explore the areas of positive Pearson correlation highlighted in our mean-filtered matrix from residue 200 to 250 (Fig 3D). This region is sparsely sampled by random insertions with the transposon-based method, and is not distinguished by crystal structure ΔDihe, so could provide previously unknown sensors.

Sequences were generated for the insertion of cpGFP into MBP every three residues in clusters 205–217 and 224–233. Sensors will be denoted by "MBP_$i$-cpGFP", where $i$ is the residue index of the insertion. For linkers, we emulated the design of Nadler *et al.* [12] (nucleotide and amino acid sequences in S1 Text). The resulting sequences were cloned into pET-28a backbone. For negative controls, cpGFP was inserted at regions of flat Pearson space: **MBP_131-cpGFP** and **MBP_195-cpGFP**. For positive controls, two very good sensors from Nadler *et al.* were reproduced: **MBP_170-cpGFP** and **MBP_348-cpGFP** [12].

Plasmids for expression of these sensors were transformed into BL21 *E. coli* and induced with IPTG. Pellets were recovered and lysed. Lysates were mixed with a pseudocytosol to simulate eukaryotic cellular conditions. The pseudocytosol was supplemented with either 0 mM, 1 mM, or 100 mM maltose, and the samples were tested for cpGFP fluorescence on a plate reader. Sensors were assayed by two metrics: raw fluorescence intensity (arbitrary units) and $\Delta F/F_0$ (where $\Delta F$ is the difference in raw fluorescence intensity between 0 mM maltose and the

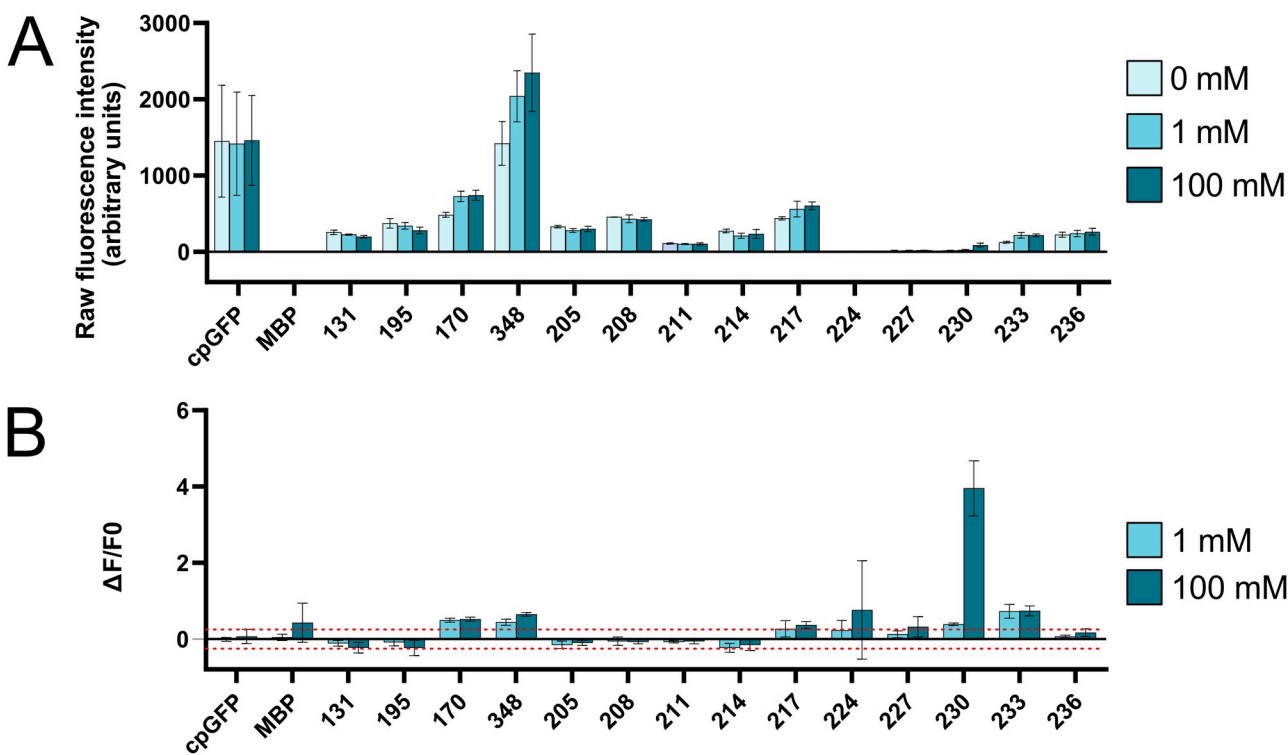

**Fig 4. Comparison of fluorescence of designed sensors at varied maltose concentrations.** On the left of the axes are the control conditions. cpGFP: the fluorescent protein alone. MBP: the Maltose Binding Protein alone. 131 and 195: cpGFP inserted at MBP's positions 131 and 195, which are both in areas of flat Pearson space so are predicted to generate poor quality sensors. 170 and 348: cpGFP inserted at MBP's positions 170 and 348, both very strong sensors previously identified by Nadler et al. [12]. Presented are the means and standard deviations of 3 biological repeats per condition. Tables for all values are in supplement. **A)** Raw fluorescence of sensors at 0, 1, and 100 mM maltose. **B)** Relative change in fluorescence from 0 mM maltose to 1 or 100 mM, calculated by $(F_0 — F_X) / F_0$. 0.25 and -20.5 are indicated with dashed lines.

given concentration, and $F_0$ is the raw fluorescence intensity at 0 mM). $\Delta F/F_0$ is a key metric in fluorescent sensor design, with an absolute value of $> 0.25$ being considered a successful design [2]. The absolute $\Delta F/F_0$ values for our negative controls **MBP_131-cpGFP** and **MBP_195-cpGFP** did not exceed 0.25.

By this standard, and when compared to previously discovered sensors, our method has generated several new viable maltose sensors (Fig 4, and S1 and S2 Tables). For example, **MBP_233-cpGFP** had lower raw fluorescence than those of Nadler's **MBP_170-cpGFP** and **MBP_348-cpGFP**, but in terms of $\Delta F/F0$ **MBP_233-cpGFP** exhibited higher values under our experimental conditions. At concentrations of 1 and 100 mM, the $\Delta F/F0$ values of **MBP_233-cpGFP** were 0.729 and 0.741, respectively, while for **MBP_170-cpGFP** they were 0.495 and 0.523, and for **MBP_348-cpGFP** they were 0.445 and 0.650. By the ">0.25" standard, **MBP_233-cpGFP** is certainly a viable sensor.

**MBP_230-cpGFP** is another sensor of note, with low raw fluorescence but by far the greatest $\Delta F/F_0$. It exhibited very little fluorescence at 0 and 1 mM and was only slightly brighter at 100 mM, with raw measured values of 22, 31, and 72 respectively. However, this small change resulted in a 100 mM $\Delta F/F_0$ of 3.954, by far the largest of those we measured. **MBP_230-cpGFP**'s fluorescence uptick at 100 mM was extremely consistent between experimental repeats, not a result of small fluctuations around a small value, making it another viable sensor, particularly for use in conditions of low background fluorescence. **MBP_227-cpGFP** also achieved $\Delta F/F_0 > 0.25$ for 100 mM, and **MBP_217-cpGFP** did so at 1 and 100 mM, totalling 4 identified viable sensors.

Not all designs were successful. **MBP_224-cpGFP** and **MBP_227-cpGFP** had no fluorescence over all conditions (the apparent large $\Delta F/F0$ for **MBP_224-cpGFP** at 100 mM is an artefact of small fluctuations). Insertions from 205–214 did not provide viable sensors, with small if any $\Delta F/F_0$ values. Clearly yet further factors than Pearson hotspots determine whether an insertion is functional or not. That said, the aim of our work was not to generate new maltose biosensors, rather it was to develop a computational method for sensor discovery that reconciles observations of previous works in the field. The method's identification of all previously known hotspots is alone proof of its success, and the 4 new viable sensors discovered in previously unknown hotspots are only further proof. Although our discovered sensors are less bright than those of previous studies, brightness is one feature that can be relatively easily improved by directed evolution if one desired to truly apply them.

## Conclusion

Modulation of effector domain output upon substrate binding is central to biosensor design. Early PBP-biosensor studies inserted effectors at regions of high backbone dihedral angle change (high $\Delta$Dihe), and attributed output modulation to direct transmission of backbone conformational changes from detector to effector. However, random insertion studies have shown that some insertions at sites of low $\Delta$Dihe give substantial substrate-induced output changes, showing that $\Delta$Dihe the insertion sites that best modulate effector output are sometimes overlooked by this metric. Here we use Pearson correlation coefficient of residue contacts in MD simulations to show that regions of significant contact change during PBP opening correlate well with the known best insertion sites. We then designed new sensors from insertions in regions of high Pearson correlation coefficient and low $\Delta$Dihe, validating this as a new method to generate PBP-biosensors.

In this work we have demonstrated that Pearson hotspots correlate with known good insertion sites from more regions of primary sequence better than changes in dihedral angles of crystal structures or MD-derived poses. The Pearson correlation coefficient matrix can be accessible with knowledge of only the *holo* crystal structure, and could even be accessed for PBPs with no known crystal structure thanks to structural prediction algorithms such as AlphaFold2, which are known to mostly predict *holo* states of binding proteins [28]. A wider variety of candidate insertion sites will help with the successful insertions of a wider variety of effectors such as transcription factors and enzymes [18,19]. It would also be interesting to see this method applied to other sensor proteins with similar hinge-like binding dynamics, such as the *de novo* hinge proteins recently developed in the Baker lab [29]. Finally, the most immediate impact of this work is the forging of a new, computational avenue for PBP-biosensor discovery that can be performed at a much greater scale than current wet-lab methods for minimal labour and cost.

## Methods

### MD simulations

Simulations were run on a computer with a 36 core CPU clocked at 4.2 GHz, 64GB RAM, NVIDIA Quadro RTX 8000 48GB VRAM, 1 TB NVMe, and 2TB SSD. Simulations were set up using AmberTools and performed using the Python package OpenMM [30]. Nonbonding interactions were modelled by PME, with cut-off at 1 nm. Simulations were run at 1 bar with a Monte Carlo barostat and 300 K temperature was maintained by the Langevin integrator with frictional constant 1 ps$^{-1}$. Hydrogen bond length constrains were applied. A timestep of 2 fs was used. Code for running these simulations can be found at https://github.com/wells-wood-research/oshea-j-wood-c-pbp-design-2023.

### RMSD, RMSF, dihedral torsion

RMSD and RMSF calculations were performed using the Python package MDAnalysis [31]. Dihedral torsion angles were calculated using the Python package MDTraj [32]. Code for calculating the change in RMSF and dihedral torsion angles can be found at https://github.com/wells-wood-research/oshea-j-wood-c-pbp-design-2023.

### CONtact ANalysis (CONAN)

CONAN was applied to the 8/10 *apo* simulations where MBP opening occurred. We defined a "contact" as any residue atoms that came within 0.5 A and then remained within 0.8 A of each other, as this is well within the limit to exclude water (which has a radius of 1.9 A). The input file and outputs relevant to this work can be found at https://github.com/wells-wood-research/oshea-j-wood-c-pbp-design-2023. For each contact in the Pearson correlation coefficient matrix, the pixel's mean value across the 8 simulations analysed was taken. These means formed the matrix that was subjected to pixel clustering, the code for clustering was adapted from that of Galloway *et al*. [33] to differentiate between peaks and valleys. For each cluster, mean pixel value, median pixel value, and maximum absolute pixel value was calculated. Peaks and valleys were then filtered for those that were in the >99th percentile of all of these values (peaks and valleys ranked separately).

### Design of constructs for expression of sensors

For a given insertion point, linker design followed that of Nadler *et al*. [12]. N-terminal linkers were $res^{i+1}$-Ala-Ser, where $res^{i+1}$ is the residue after the inserted residue. C-terminal linkers were Ala-Ser-$res^i$, where $res^i$ is the residue that was inserted at. The amino acid sequence of cpGFP flanked by the linkers was inserted into the amino acid sequence for MBP. The amino acid sequence was codon optimised for expression in *Escherichia coli* using the Twist Bioscience sequence input interface. Optimised sequences were cloned into the pET28a backbone at the SacI_HindIII restriction sites.

### Cloning of constructs

Primers for cloning and sequencing were ordered from IDT. Fragments were amplified using Phusion polymerase (NEB), run on 1% agarose gels and recovered using GeneJET Plasmid miniprep kit (ThermoFisher). Recovered fragments were assembled using NEBuilder kit (NEB). Assembled plasmids were sequenced by the Medical Research Council Protein Phosphorylation and Ubiquitylation Unit's DNA sequencing service at the University of Dundee.

### Sensor expression

Plasmids were transformed into BL21-alpha cells (NEB) and plated overnight at 37˚C on kanamycin LB plates (50 μg/mL). The next day, colonies were picked into 2 mL of TB supplemented with kanamycin (50 μg/mL), and incubated at 37˚C, 190 rpm for 20 hours. 500 uL of these cultures were then used to inoculate 25 mL of LB supplemented with kanamycin (50 μg/mL), and incubated at 37˚C, 240 rpm for 1.5 hours. Cultures were then supplemented with 1 mM IPTG, and further incubated at 25˚C, 240 rpm for 20 hours.

### Fluorescent assay

After expression, cultures were centrifuged at 4000 G, 4˚C for 15 minutes. The pellets were recovered and weighed, then resuspended in 1 uL/mg of lysis buffer (1x BugBuster (Novagen), 20 mM Tris-HCl, 150 mM K-gluconate, pH 7.5 [34]). Lysis was performed with shaking at

room temperature, 1600 rpm, 30 minutes. Lysates were spun down at 16000 G for 15 minutes and the supernatant was recovered. 20 uL of supernatant was transferred to wells in the 96 well black/clear bottom plate (ThermoFisher). 200 uL of pseudocytosol (100 mM K-gluconate, 30 mM NaCl, 25 mM MES, 25 mM HEPES, 40% sorbitol, 1mg/mL bovine serum albumin, pH 7.5 [34–36]) supplemented with 0, 1, or 100 mM maltose. Plates were immediately imaged on an Infinite M200PRO (Tecan) plate reader, with excitation/emission of 485/515nm, with gain set to 40. In each of the three biological repeats, three technical repeats were performed for each condition and from these the mean was calculated. The mean value of the background negative control pET28a was subtracted from the positive controls and experimental conditions. From this background-subtracted mean value, $\Delta F/F_0$ values were calculated, where $\Delta F = F_X - F_0$, where X = 1 or 100. For the MBP negative control, 5 µg of MBP in 5 µL (RayBiotech) was mixed with 15 µL of lysis buffer and 200 µL of pseudocytosol with 0, 1, or 100 mM maltose.

## Supporting information

**S1 Fig. RMSD of the *apo* and *holo* simulations with respect to the maltose binding protein's *apo* crystal structure (PDB: 1OMP).**
(TIFF)

**S2 Fig. Unfiltered matrix of mean Pearson Correlation Coefficient values for inter-residue contacts over all apo simulations where the closed-to-open transition occurred.**
(TIFF)

**S3 Fig. Pearson Correlation Coefficient matrix of inter-residue contacts calculated from comparison of the available crystal structures of apo (1ANF) and holo (1OMP) MBP.**
(TIFF)

**S1 Table. Mean and individual raw fluorescence intensity values for all sensors and controls.** "Replicate" refers to "biological replicate", each being a mean of three technical repeats.
(XLSX)

**S2 Table. dF/F0 values for all sensors and controls.**
(XLSX)

**S1 Text. FASTA format nucleotide and amino acid sequences used in this study.** Maltose binding protein sequence (MBP) is highlighted in blue, circularly permuted GFP sequence (cpGFP) in yellow, and linker sequence between these two components in biosensors in green.
(DOC)

## Acknowledgments

Thank you to Dr. Frank Machin of the Doerner lab for direction to literature and Jonathan Lecoy of the Richardson lab for help using their lab space. Thank you to the teams at the Edinburgh Genome Foundry and the Edinburgh Protein Production Facility.

## Author Contributions

**Conceptualization:** Jack M. O'Shea, Peter Doerner, Annis Richardson, Christopher W. Wood.

**Data curation:** Jack M. O'Shea.

**Formal analysis:** Jack M. O'Shea.

**Funding acquisition:** Peter Doerner, Annis Richardson, Christopher W. Wood.

**Methodology:** Jack M. O'Shea, Christopher W. Wood.

**Resources:** Peter Doerner.

**Software:** Jack M. O'Shea, Christopher W. Wood.

**Supervision:** Peter Doerner, Annis Richardson, Christopher W. Wood.

**Validation:** Jack M. O'Shea.

**Visualization:** Jack M. O'Shea.

**Writing – original draft:** Jack M. O'Shea, Peter Doerner, Christopher W. Wood.

**Writing – review & editing:** Jack M. O'Shea, Peter Doerner, Annis Richardson, Christopher W. Wood.

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
