## [Decision Letter · Decision Letter 0]

20 Feb 2024

Dear Dr Wood,

Thank you very much for submitting your manuscript "Computational Design of Periplasmic Binding Protein Biosensors Guided by Molecular Dynamics" for consideration at PLOS Computational Biology.

As with all papers reviewed by the journal, your manuscript was reviewed by members of the editorial board and by several independent reviewers. In light of the reviews (below this email), we would like to invite the resubmission of a significantly-revised version that takes into account the reviewers' comments.

We cannot make any decision about publication until we have seen the revised manuscript and your response to the reviewers' comments. Your revised manuscript is also likely to be sent to reviewers for further evaluation.

Sincerely,

Alexander MacKerell

Academic Editor

PLOS Computational Biology

Nir Ben-Tal

Section Editor

PLOS Computational Biology

Reviewer's Responses to Questions

**Comments to the Authors:**

Reviewer #1: O’Shea et al. has demonstrated how investigation of protein dynamics provides useful information that can be used to design fluorescence based biosensors. While the overall results are convincing and supported by experimental results, I believe a few more analysis can greatly improve the quality of the manuscript.

1. On page 8, the author writes "we ran 10 x 200 ns of these apo simulations, observing closed-to-open transition in 8/10 of them,". It is a bit vague what criteria authors have used to make this decision. Please provide a quantitative description to support this statement.

2. In Figure 2, panel E, caption, the authors write that they show a "representative" model for open or closed state. How the model was selected, please provide a quantitative description.

3. In figure 2, authors show RMSD compared to the starting structure. How does the RMSD looks like compared to the open state?

4. The manuscript seems to be missing a "Conclusion section". Please provide one.

Reviewer #2: The paper suggests that residue position can add to the insights from backbone dihedral angles to generate additional PBP-based biosensors. However, the paper’s major data point is suspect, and the presentation needs improvement.

Data:

Experiments: one week’s experiments would probably help. Especially for residue 230—the star of the paper-- the paper’s point is suspect because (a) the counts are so small, and (b) the only increase measured occurs at 100 mM—roughly 100-fold greater than precited from Ref. 10. Because 230 is in the binding pocket for maltose (again, ref (10)), one suspects that mutating this residue violates the authors’ “agnostic” principle—which was never intended to extend to the substrate binding site. I suspect that authors have constructed a defective binding site. It will be necessary to purify the protein—a simple task if the construct includes a His6 sequence. A full dose-response relation should follow.

Explanations:

The paper ought to better explain the calculations that underlie the suggestions for experiments. reader The final intelligible sentences is ‘We defined a “contact” . . . .’ After that, the become mysterious to the non-cognoscenti. The mysterious concepts include “pixel”, “pixel clustering”, and the residue-level metrics being correlated with each other. Are you simply asking which residues move together (positive) and which move separately? It would be good to say so. GITHUB is not peer-reviewed, so we don’t whether its explanations are clear.

It would also be good to state the hardware used to perform he simulations and to analyze the data, so the reader knows whether (in 2024) to perform the analyses using a phone, a tablet, a laptop, a desktop, or a supercomputer.

Additional confusion: the matrix of 3D is symmetrical, but the figure does not state this and shades only along the y-axis. And the correlation color map is puzzlingly congruent with the Y-axis. If the authors give though to the poor reader, this would solve many of the paper’s problems in presentation.Some of the paper’s puzzling features would improve if the experimentalists and computationalists simply talk with each other. Thus, what some of the authors call “raw fluorescence”, others call F0. What one set of authors called “uptick” another set would term, “dose-response relation”.

Pedantic corrections:

“The Pearson Correlation Coefficient matrix can be accessed from only the holo crystal structure,”. Is a very confusing statement, in part because “only” has many interpretations. Perhaps it means, “Knowledge the holo structure is sufficient to begin MD simulations, without the necessity for knowing the apo structure.”

Here is the correct capitalization: Person correlation coefficient. Pearson is a proper noun; his correlation coefficient is not. Although one might capitalize the abbreviation (PCP), one does not capitalize the individual words.

**Have the authors made all data and (if applicable) computational code underlying the findings in their manuscript fully available?**

Reviewer #1: **No: **Please upload the raw simulation trajectories to any open source data sharing services.

Reviewer #2: Yes

PLOS authors have the option to publish the peer review history of their article (what does this mean?). If published, this will include your full peer review and any attached files.

Reviewer #1: **Yes: **Nandan Haloi

Reviewer #2: No
---

## [Decision Letter · Decision Letter 1]

27 Apr 2024

Dear Dr Wood,

Thank you very much for submitting your manuscript "Computational Design of Periplasmic Binding Protein Biosensors Guided by Molecular Dynamics" for consideration at PLOS Computational Biology. As with all papers reviewed by the journal, your manuscript was reviewed by members of the editorial board and by several independent reviewers. The reviewers appreciated the attention to an important topic. Based on the reviews, we are likely to accept this manuscript for publication, providing that you modify the manuscript according to the review recommendations.

Sincerely,

Alexander MacKerell

Academic Editor

PLOS Computational Biology

Nir Ben-Tal

Section Editor

PLOS Computational Biology

Reviewer's Responses to Questions

**Comments to the Authors:**

Reviewer #1: The authors have fully addressed the concerned raised.

Reviewer #2: This critique repeats some of my original comments. Please take additional care to respond to these comments. Part of our profession is efficient communication. At present, some aspects require the reader to work unnecessarily hard.

Figure 3D still needs work. At present, it’s a square. It should be simply an equilateral right triangle (the lower right half of the present square). The suggested tactic would eliminate the redundant, confusing part. The suggested tactic would eliminate the necessity to explain (to stupid readers like me) why the shaded vertical bars are not also shaded horizontal bars. Again, the fact that the color lookup table has the same height as the vertical axis is also confusing. Please make the color lookup table smaller than the vertical axis, so that harried readers (like myself) don’t have to ask whether the lookup table encodes the vertical axis (it does not!).

Th Journal’s web site does not allow me to view the supplementary material. Therefore, I cannot tell whether the authors have rectified the “F0” vs “raw fluorescence” terminology.

FWIW, what’s the cost of the computer time, the time awaiting intermediate steps in the computational pipeline, and the like? Compared with the cost of time and materials for one round of site-saturated mutagenesis? Is this comparison not even worth considering?

**Have the authors made all data and (if applicable) computational code underlying the findings in their manuscript fully available?**

Reviewer #1: None

Reviewer #2: Yes

PLOS authors have the option to publish the peer review history of their article (what does this mean?). If published, this will include your full peer review and any attached files.

Reviewer #1: **Yes: **Nandan Haloi

Reviewer #2: No

Figure Files:

Data Requirements:

Reproducibility:

References:

---

## [Editor Report · Decision Letter 2]

30 May 2024

Dear Dr Wood,

We are pleased to inform you that your manuscript 'Computational Design of Periplasmic Binding Protein Biosensors Guided by Molecular Dynamics' has been provisionally accepted for publication in PLOS Computational Biology.

Best regards,

Alexander MacKerell

Academic Editor

PLOS Computational Biology

Nir Ben-Tal

Section Editor

PLOS Computational Biology

---

## [Editor Report · Acceptance letter]

12 Jun 2024

PCOMPBIOL-D-24-00104R2 

Computational Design of Periplasmic Binding Protein Biosensors Guided by Molecular Dynamics

Dear Dr Wood,

I am pleased to inform you that your manuscript has been formally accepted for publication in PLOS Computational Biology. Your manuscript is now with our production department and you will be notified of the publication date in due course.

With kind regards,

Olena Szabo
